# An artificial chromosome ylAC enables efficient assembly of multiple genes in *Yarrowia lipolytica* for biomanufacturing

Zhong-peng Guo [1✉], Vinciane Borsenberger[1], Christian Croux[1], Sophie Duquesne[1], Gilles Truan[1], Alain Marty[1] & Florence Bordes[1]

The efficient use of the yeast *Yarrowia lipolytica* as a cell factory is hampered by the lack of powerful genetic engineering tools dedicated for the assembly of large DNA fragments and the robust expression of multiple genes. Here we describe the design and construction of artificial chromosomes (ylAC) that allow easy and efficient assembly of genes and chromosomal elements. We show that metabolic pathways can be rapidly constructed by various assembly of multiple genes in vivo into a complete, independent and linear supplementary chromosome with a yield over 90%. Additionally, our results reveal that ylAC can be genetically maintained over multiple generations either under selective conditions or, without selective pressure, using an essential gene as the selection marker. Overall, the ylACs reported herein are game-changing technology for *Y. lipolytica*, opening myriad possibilities, including enzyme screening, genome studies and the use of this yeast as a previous unutilized bio-manufacturing platform.

[1] TBI, Université de Toulouse, CNRS, INRA, INSA, Toulouse, France. ✉email: zguo@insa-toulouse.fr

   

Alone at the bottom of the hemiascomycetous yeast tree, *Yarrowia lipolytica* is phylogenetically distant from the industrial work-horse *Saccharomyces cerevisiae* and all other well-studied yeast species[1]. This unconventional obligate aerobe, can feed on a wide range of carbon sources (including lipids, n-alkanes and glycerol), accumulate high cellular content of lipids and secrete large amounts of organic acids, such as citric acid, isocitric acid, and α-ketoglutaric acid[2,3] and even proteins[4]. For decades, these desirable physiological traits and its "generally recognized as safe" (GRAS) status[5], have made *Y. lipolytica* a strain of interest for the detergents, food, pharmaceutical and environmental industries[4,6,7]. Inevitably, industrial interest in *Y. lipolytica* has provided the impetus to develop a variety of basic genetic tools, including those for protein expression[8–12], YaliBrick-based cloning[13], iterative gene integration[14] and nuclease-based genome-editing[15–18], etc. Combined with the release of the full genome sequence of *Y. lipolytica* (CLIB 122) in 2004[19], these tools fuelled basic and applied research that has yielded modified strains capable of producing biofuels, pharmaceuticals, basic commodities and building blocks for the chemical industry[20–22]. Moreover, engineering of the *Y. lipolytica* lipid biosynthesis pathway has provided the means to produce a variety of products, including fatty acid ethyl esters, fatty alkanes, medium chain-length fatty acids, polyunsaturated fatty acid (e.g. eicosapentaenoic acid), fatty alcohols and triacylglycerides (TAGs), all of which being obtained through single or multiple gene insertions[23–26].

Despite these achievements, building complex pathways in *Y. lipolytica* remains challenging. Multiple gene integration is time-consuming, since it requires repeated transformation-selection cycles and the rescue of selection markers[21]. Likewise, sequential genome insertions are generally hampered by decreasing transformation efficiencies and progressive loss of strain fitness[27]. Finally, non-homologous end-joining (NHEJ) is predominant over homologous recombination (HR)[28]. Overall, site directed integration of DNA into the *Y. lipolytica* genome is inefficient and requires large (~ 1 kb) target sequences bearing 5′- and 3′-flanking regions[28,29]. With an average transformation efficiency of 2500 cfu/μg, HR yields only 1% of positive transformants. To overcome assembly and integration problems, various methods which form the cornerstone of modern synthetic biology for *S. cerevisiae*[30–32] were proposed for *Y. lipolytica*. These include DNA assembler (four genes, ~11 kb, assembling efficiency 20%)[33] and Golden Gate Assembly (four genes, ~11 kb, assembling efficiency ~70%)[34]. Unfortunately, due to the different inherent limitations between the two yeast species, the number of genes that can be introduced and length of the assembled DNA both remain limited for *Y. lipolytica*. Finally, it is noteworthy that random integration of expression cassettes, a method that is frequently preferred over site specific integration, often leads to locus-dependent gene expression[35] and even sometimes unpredictable phenotypes[36]. To improve HR efficiency, researchers have explored the inactivation of the NHEJ pathway[28,37]. While this strategy increases HR frequency[28], NHEJ pathway inactivation provokes a strong negative impact on the robustness and fitness of *Y. lipolytica*, in particular regarding stress resistance[38] and transformation efficiency[28].

The recent introduction of nuclease-based strategies (e.g. Cas9, Cas12a/Cpf1 and TALEN)[16–18,39] has provided the means to overcome some of the above-mentioned limitations. However, HR still relies on the use of large homology regions of 1 kb or more to achieve satisfactory recombination efficiencies[40]. Moreover, both the integration yield[35] and the heterologous gene expression levels[41] remain locus dependent. All these are drawbacks contribute to the continued preference of researchers for *S. cerevisiae* despite the advantageous metabolic capabilities of *Y. lipolytica*[42,43].

Yeast artificial chromosomes (YACs) are shuttle (*E. coli* and *S. cerevisiae*) vectors containing autonomously replicating sequences (ARS), one centromere (CEN) and two telomeric elements (TEL)[44]. They were originally dedicated systems for *S. cerevisiae* and have historically been used for cloning and manipulating large DNA inserts such as full-scale genomic libraries for mapping and functional analysis purposes[45,46]. In this study, we have designed and developed a completely synthetic *Y. lipolytica*-specific artificial chromosome (ylAC). Herein, we describe the construction and use of ylAC to introduce a complete cellobiose and xylose co-utilization pathway in *Y. lipolytica*.

## Results

**Design and construction of ylAC systems for *Y. lipolytica*.** To mimic the native structure of genomic DNA, we constructed a ylAC that contains the essential replication and segregation elements of *Y. lipolytica* chromosomes (Fig. 1). Instead of using ligation based YAC construction as described previously[44], we investigated whether the genes could be directly assembled in vivo using HR of *Y. lipolytica*. As a proof-of-concept, we developed a workflow (Fig. 2a) in which DNA assembly, replication of the construct and gene expression are performed simultaneously.

As repeated sequences are often the source of YAC instability, a panel of promoters and terminators were characterized. The *RedStar2* reporter gene was used to evaluate the strength of different endogenous *Y. lipolytica* promoters, including previously identified EXP, FBA, GPAT and TDH[47], and uncharacterized ones (GPD, GUT, HXK1, HXK2, HXK3, PGK and TPI) pairing with their corresponding terminators (Supplementary Table 1). TEF promoter and LIP2 terminator were used as control for comparison. The transformants expressing *RedStar2* under the control of different promoters and terminators showed similar growth rates and substrate consumption rates on YPD media (Supplementary Fig. 1). The fluorescence intensity of the cell suspensions was measured during the cell cultures. The fluorescence intensity was first divided by the optical density (OD) of the sample and then normalized by the highest value, in this case, under the control of TEF promoter and LIP2 terminator. Based on this analysis, the relative order of promoters' strengths was TEFå TDHå EXPå FBAå GUTå TPI (Supplementary Fig. 1). The strengths of all other promoter/terminator pairs were less than 10% of that of the pTEF/tLIP2.

In a first instance, a simple, restriction enzyme-digested ylAC assembling with two reporter genes, *RedStar2* (encoding DsRed fluorescent protein) and *EG2* (encoding endoglucanase) was used to evaluate the importance of ARS and TEL sequences on the efficiency of DNA assembly. Using this ylAC to transform *Y. lipolytica* Po1d revealed that the absence of ARS (TEL⁻ARS⁻ and TEL⁺ARS⁻) resulted in very low (≥11 cfu/μg DNA for TEL⁺ and ≥100 cfu/μg DNA for TEL⁻) transformation efficiency. Inversely, in the presence of ARS (TEL⁻ARS⁺ and TEL⁺ARS⁺), transformation efficiency was increased by two orders of magnitude (2500–3000 cfu/μg DNA), thus revealing that ARS is an essential factor to obtain high transformation efficiency with ylAC (Fig. 2b). On the contrary, the presence of TEL has little or no impact on transformation efficiency, although it improves the assembly yield (i.e. the % of colonies that bear the full DNA theoretical construct). In the presence of ARS alone (TEL⁻ARS⁺), assembly yield only reached 20%, but was increased to 60% when both ARS and TEL were present. Consistent with a higher assembly yield, more than 60% of the TEL⁺ARS⁺ transformants displayed both EG2 activity and DsRed fluorescence (Fig. 2c). However, in the absence of ARS (TEL⁻ARS⁻ and TEL⁺ARS⁻) the number of colonies displaying these functions was reduced by 50% (Fig. 2c). In addition, none of these colonies contained the

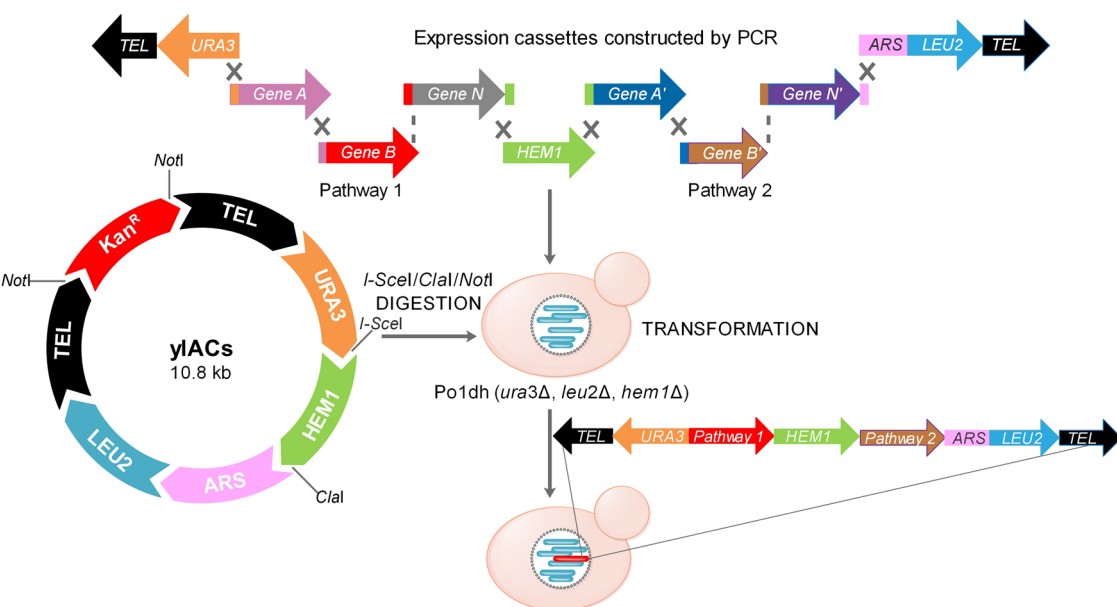

**Fig. 1 Construction ylAC, a *Y. lipolytica*-specific artificial chromosome.** To ensure autonomous replication and segregation during cell division, ylAC is composed of two telomeres (TEL), an autonomous replication sequence (ARS), which contains a centromere (CEN3-1) (12) and an origin of replication. The digestion of ylAC using *Not*I, *I-sce*I and *Cla*I generates three linear fragments displaying telomeric sequences at both extremities. Each fragment contains different selection markers (*URA3* on the left arm, *HEM1* gene in the middle, *LEU2* fused with ARS on the right arm). The full assembly of ylAC modules is achieved by HR in *Y. lipolytica* using the digested ylACs plasmid and PCR-amplified expression cassettes.

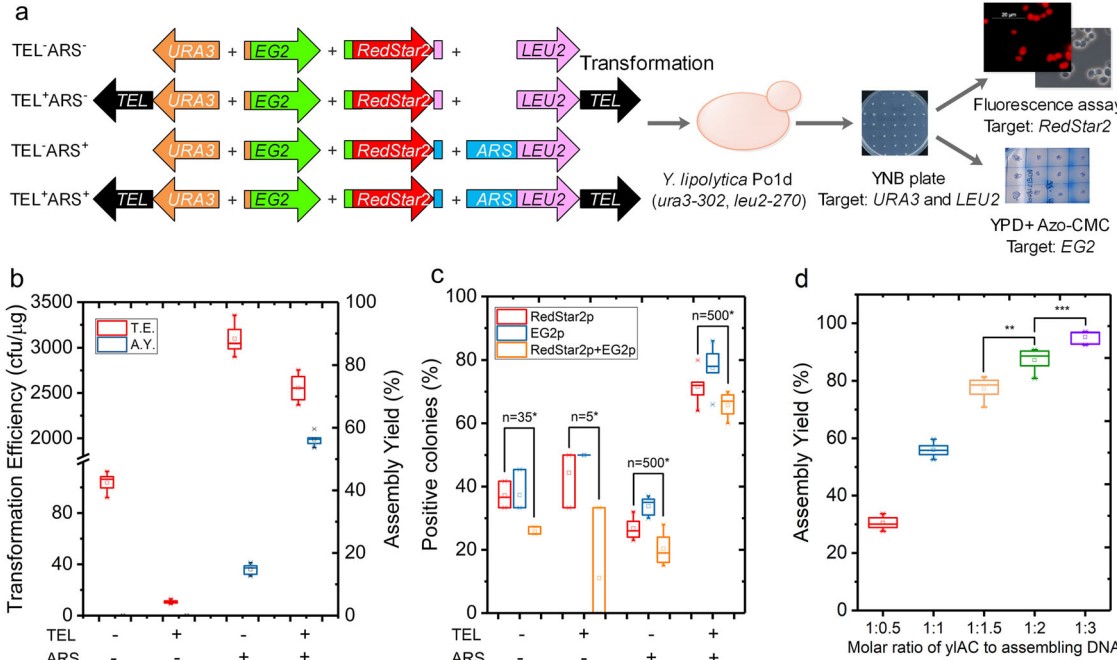

**Fig. 2 Both ARS and TEL are necessary for efficient and correctness of the DNA assembly. a** Experimental workflow. Different combinations of chromosomal elements (ARS and TEL) are assembled with expression modules and transformed into *Y. lipolytica* Pod1d. Each module contains 50-bp flanking regions homologous to its adjacent sequence. Transformants selected for auxotrophic markers are then tested for the production of the reporter proteins. **b** Comparison of the Transformation Efficiency (TE) and Assembly Yield (AY) using different combinations of ARS and TEL. TE was calculated by dividing the number of successful transformants by the amount of DNA used during a transformation procedure. AY was calculated by dividing the number of transformants harbouring the expected assemblies by the total number of the transformants verified. **c** the percentage of transformants expressing either or both phenotypes (fluorescence and endoglucanase activity) within the total cell population (*n* represents the number of obtained clones). Asterisk indicates that none of them contained the complete gene-assembly. **d** AY as a function of the molar ratio of expression modules and ylAC elements (ARS and TEL). (**P value < 0.05, ***P value < 0.01, two-tailed Student's *t*-tests). The results were calculated from at least five biological replicates and are given as the mean value and standard deviation.

complete gene assembly and these few transformants correspond to chromosomal integration of the expression modules (Fig. 2b; Supplementary Fig. 2). Transformation of the yeast cells without any DNA or with the target genes but without selection markers did not yield any colonies.

To optimize DNA assembly, the ratio of expression modules and ylAC elements (ARS and TEL) was varied. Assembly yield increased when the proportion of the ylAC was decreased with respect to that of the expression modules, reaching 95% for a ratio of 1:3 (ylAC:expression modules) (Fig. 2d), while transformation efficiency remained unchanged (2500 ± 62 cfu/µg DNA). Thereafter, a 1:3 ratio was employed to ensure maximal assembly yield.

**Episomal status of the artificial chromosome.** To examine whether ylAC constructs adopt a chromosome-like structure, a Southern blot analysis was performed on genomic DNA (gDNA) using a *RedStar2*-specific probe, first ensuring that the gDNA is representative of cells bearing the correct ylAC assembly (Supplementary Fig. 2). DNA sequence analysis revealed that ARS, TEL and the reporter genes were all present (e.g. YTA1), and the *Cla*I/*Nde*I-digestion of the gDNA yielded the two expected DNA fragments (i.e. 3.0 and 4.5 kb, respectively), while *Cla*I-digestion only produced a single fragment exhibiting a length of 7.4 kb (Fig. 3b, lanes 4, 5 and 7, 8; Supplementary Fig. 3). Additionally, a single species exhibiting an approximate size of 9.4 kb was observed for the undigested gDNA, consistent with the expected linear length of the complete ylAC-EG2-RedStar2 construct (ylAC-ER). On the other hand, when gDNA of WT *Y. lipolytica* was analysed, no restriction fragments were detected (Fig. 3), and when the *RedStar2* gene was integrated into the genome, the gDNA of the transformants also displayed different profiles (e.g. TEL⁻ARS⁻ and TEL⁺ARS⁻, Supplementary Fig. 4).

The exclusion of TEL from the ylAC gene constructs (e.g. YA1) yielded transformants bearing circular plasmids (Supplementary Fig. 4). This conclusion was reached using Southern blot analysis revealed that undigested YA1 gDNA migrates as a single species that exhibits an apparent molecular size inferior to that of the DNA digested with *Cla*I. This infers that the undigested YA1 gDNA is a covalently closed circular DNA molecule. Likewise, qPCR analysis performed on YTA1 revealed the presence of 1 to 2 copies of ylAC, while a similar analysis performed on YA1 grown on selective medium revealed 2–3 copies of the circular DNA molecule (Supplementary Table 2).

The transferability of the ylAC gene assemblies was examined by transforming Po1d with gDNA from YTA1 (linear) and YA1 (circular) and selecting for URA⁺ clones. Successfully transformed cells produced DsRed and EG2 proteins at levels comparable to the parental strains (Fig. 3c) and Southern blot analysis of their gDNA (e.g. from YTA1-1 and YA1-1) revealed profiles identical to those of YTA1 and YA1 respectively (Fig. 3b; Supplementary Fig. 4). Overall, these results confirm that the in vivo assembly method used to obtain the ylAC gene constructs is adequate to obtain independent chromosomal entities that fulfil all of the functions (i.e. autonomous replication, cell segregation, gene expression etc) that are associated with native *Y. lipolytica* chromosomes.

**Enhancing the construction efficacy and stability of ylAC.** Comparing the stability of CEN plasmids and integrated genes under non-auxotrophic selection conditions has revealed that gene integration provides higher stability in *S. cerevisiae*[48]. Consequently, we investigated the mitotic stabilities of circular and linear ylAC gene assemblies in *Y. lipolytica*. Both linear and circular assemblies were stable when selective pressure was applied, with typical losses of the auxotrophic marker being lower than 0.07% per generation (Table 1). Moreover, 98% of the cells contained the ylAC gene assemblies even after 30 generations, this being irrespective of the ylAC form (i.e. linear or circular). However, in the absence of auxotrophic selective pressure, both linear and circular ylAC gene assemblies were lost at substantially higher rates (~6% loss per generation) and about 80% of cells had lost the ylAC gene assemblies after 30 generations.

Natural chromosomes are stable in most conditions, irrespective of the ambient culture conditions. Therefore, to improve the stability of the linear (chromosomal) form of ylAC in *Y. lipolytica*, we designed a maintenance strategy that ensures chromosomal stability irrespective of whether selection pressure is applied or not. To achieve this, *HEM1*, which encodes the essential 5-aminolevulinate synthase function, was included in the ylAC as a new selection marker, yielding ylAC2. Reciprocally, using CRISPR/Cas9 a *HEM1* gene deletion was created in the genome of *Y. lipolytica* Po1d strain, yielding the 5-aminolevulinate synthase defective strain Po1dh (Supplementary Fig. 5). To propagate the Po1dh strain, 5-aminolevulinate was added to the culture medium.

To evaluate the complementation system, a ylAC2 construction encoding *RedStar2* and *EG2* was assembled and used to transform *Y. lipolytica* Po1dh using the previously established method (Fig. 1). Notably, transformation efficiency was decreased by 30%

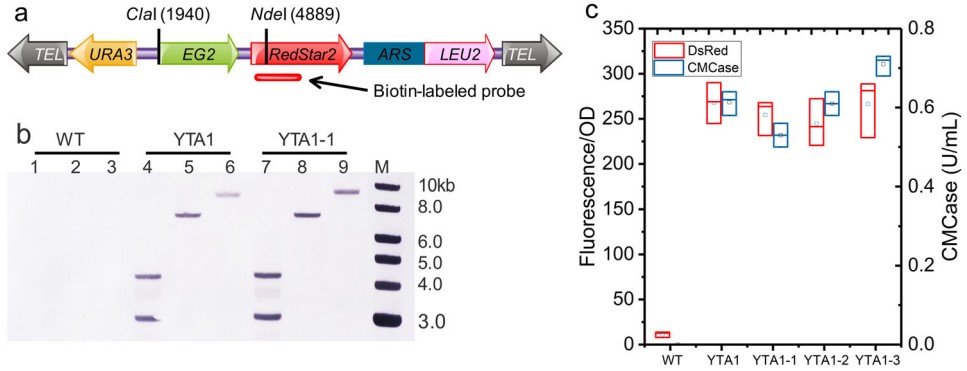

**Fig. 3 The ylAC gene assemblies are extrachromosomal, replicative and transferable genetic entities. a** Configuration of the target DNA assembly, the restriction enzyme sites, and position of the DNA probe are indicated. **b** Southern blot analysis of the gDNA of the wild type strain (WT), the initial transformant (YTA1), and *Y. lipolytica* re-transformed with the linear DNA fragment isolated from YTA1 (YTA1-1, 2 and 3). The *RedStar2* gene was used as the target for a specific probe to detect gDNA fragments generated by restriction enzyme digestion. Lanes 1, 4, 7 and: gDNA digested by *Cla*I/*Nde*I (expected sizes: 4494 and 2949 bp), lanes 2, 5, 8: gDNA digested by *Cla*I (expected size: 7442 bp), lanes 3, 6, 9: undigested gDNA (expected size: 9381 bp). **c** Quantification DsRed expression by detection of fluorescence produced by transformants YTA1-1 to YTA1-4 and by WT strain.

**Table 1 Comparison of stability of linear and circular DNA assemblies of ylACs-*EG2-RedStar2* in mitotic cells of *Y. lipolytica* in aerobic cultivation with and without selection pressure.**

| | Selective condition | | Non-selective condition | |
| --- | --- | --- | --- | --- |
| | % cells with ylACs | % loss per generation | % cells with ylACs | % loss per generation |
| Random Integration | 92 ± 5 ($N \approx 30$) | 0.32 ± 0.14 | 77 ± 7 ($N \approx 30$) | 0.9 ± 0.15 |
| Circular (ylAC) | 98 ± 1 ($N \approx 30$) | 0.06 ± 0.01 | 22 ± 3 ($N \approx 30$) | 5.0 ± 0.5 |
| Linear (ylAC) | 97 ± 1 ($N \approx 30$) | 0.07 ± 0.01 | 17 ± 2 ($N \approx 30$) | 6.2 ± 0.4 |
| Linear (ylAC2) | 97 ± 1 ($N \approx 30$) | 0.05 ± 0.02 | 96 ± 1 ($N \approx 30$) | 0.5 ± 0.1 |

Results were calculated from at least five biological replicates and are given as the mean value ± standard deviation. N represents the number of generations.

compared to the previous work using ylAC, but assembly yield was increased to >90% (Supplementary Fig. 6). More importantly, as anticipated the stability of ylAC2 in non-selective conditions was significantly improved, the rate of loss being <0.05% per generation. Moreover, >98% of the cells still bore the ylAC2 gene assembly after 30 generations (compared to 22% for ylAC, Table 1). Finally, when grown in rich media, cells bearing ylAC2 exhibited similar growth and gene expression (DsRed and EG2) profiles when compared to a *Y. lipolytica* strain in which both genes were inserted in the natural chromosomes (Supplementary Fig. 7). This contrasts with the behaviour of ylAC-bearing cells, which displayed a diminished ability to produce DsRed and EG2 (Table 1; Supplementary Fig. 7). Therefore, while the presence of *HEM1* stabilizes ylAC2 during mitosis, the absence of detectable changes in growth rate also indicates that the ratio of ylAC2 to the other chromosomes is probably 1:1.

**Using ylAC for enzyme screening and pathway optimization.** Having established a straightforward, robust method for the assembly of stable, functional artificial chromosomes capable of driving gene expression, we explored the use of ylAC for large pathway engineering. To deliver a proof of concept, we decided to confer cellobiose catabolism to *Y. lipolytica*, this phenotype being the result of the action of several enzymes and transporters[6,49] (Fig. 4a).

*Y. lipolytica* strains were constructed, successively adding the components of the catabolic system. The introduction of cellobiose phosphorylase (strain ylCBP, Supplementary Table 3) provided evidence that its presence alone is insufficient to support growth on cellobiose. However, upon the introduction of CDT1 transporter (ylCello), partial (compared to growth on glucose) growth was detected. Comparing the intracellular metabolites of ylCello grown on cellobiose with those of the WT strain grown on glucose revealed that both displayed similar Glc-1P and trehalose levels, but ylCello displayed a 4 times higher glycogen level (Supplementary Fig. 8; Fig. 4b, c). This indicates that native phosphoglucomutases (PGM) in *Y. lipolytica* favors the conversion of Glc-1P to glycogen storage rather than G6P production. To address this hypothesis and redirect the flux of Glc-1P to Glc-6P, five different yeast PGMs were used to perform random in vivo assembly along with the CBP and CDT1 modules and the ylAC elements. Of the 120 clones that grew on cellobiose, a thorough analysis was performed on 30 to evaluate DNA assembly (Supplementary Fig. 9), intracellular metabolites (Supplementary Fig. 10) and growth (Fig. 4; Supplementary Table 3). The ylCello strain, expressing the *S. cerevisiae* PGM2 (an enzyme that is 2 times more active than that of *Y. lipolytica*) and growing on cellobiose, displayed a growth rate, biomass yield and glycogen content similar to that of WT strain grown on glucose (Fig. 4b). Further PCR verification, Southern blot analysis and sequencing of the gDNA (strains ASC1 and ASC2, Supplementary Figs. 11 and 12) confirmed both the successful assembly of a rather large ylAC (15 kb) containing the target genes (*CBP1, CDT1* and *scPGM2*) and its transferability (strain ASC1-1, Supplementary Fig.

12). The results thus illustrate how a modulable and efficient ylAC reconstitution system can be used for enzyme discovery and screening prior to the optimization of multi-enzyme metabolic pathways.

**The easy assembly of large metabolic pathways using ylAC.** Encouraged by the success of the previous experiment, we focused on the more ambitious goal of conferring both xylose utilization and cellobiose catabolism to *Y. lipolytica*, ensuring that glucose assimilation does not repress xylose utilization (Fig. 5). To achieve this, three key genes for xylose utilization (*XYL1, XYL2* and *XKS1*) and three for cellobiose consumption (*CBP1, CDT1* and *scPGM2*) were assembled in a 23 kb ylAC (Fig. 5a).

Transformants were first grown on YNB glucose medium and then further tested for growth on YNB xylose and YNB cellobiose media. Ninety percnetage of the colonies that grew on xylose also exhibited the ability to grow on cellobiose. Consistent with our previous observations, transformation efficiency was much higher when ARS was included in the DNA assembly, and transformation efficiency and assembly yield were high only when both of ARS and TEL sequences were present (Fig. 5b, Supplementary Fig. 13). It is remarkable that assembly yield even reached 90 % when ylAC2 was used (Supplementary Fig. 6). Analysis of gDNA of ASXC1 and ASXC2 confirmed the presence of the ylAC-*XYL1-XYL2-XKS1-CBP-CDT-scPGM2* assembly (designated ylAC-XC) and further experiments revealed that this construction can be easily transferred between strains (Supplementary Fig. 14). Strain ASXC1, containing both xylose and cellobiose utilization pathways required 36 h to completely consume xylose and glucose (Fig. 5c, d), with glucose consumption being more rapid. However, this strain consumed xylose and cellobiose simultaneously, fully depleting these carbon sources after 28 h. Remarkably, growth saturation was reached 8 h earlier that when ASXC1 was grown on a mixture of glucose and xylose. To our knowledge, this proof of concept is the first to demonstrate how *Y. lipolytica* can be metabolically engineered, introducing a large (23 kb) functionally complex DNA using a rather straightforward, robust method based on the use of a yeast artificial chromosome.

**Discussion**
Herein, we describe the design, construction and successful use of a *Y. lipolytica*-specific artificial chromosome (ylACs). The ultimate highlight of this work is the assembly of, but not limited to, a 23 kb ylAC bearing eight genes, that was introduced into *Y. lipolytica* in a single step that required less than one week of wet lab experimentation. Compared to conventional multi-step methods (Supplementary Fig. 15), this method is no doubt much faster and provide more reliable results[21,27,49,50].

Among the remarkable features of ylAC, is the fact that it only requires 50 bp homologous sequences to reach very high assembly yield (>90% for 8 genetic modules) and thus unprecedented success rates. This might be the result of the high accessibility of the ylAC

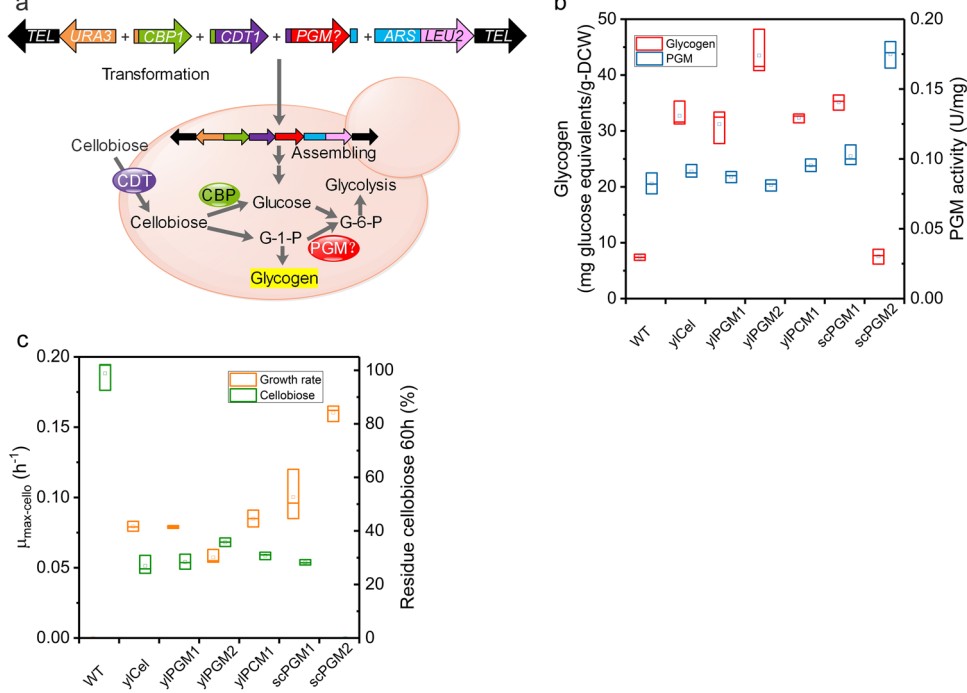

**Fig. 4 PGM screening using ylAC in *Y. lipolytica* modified strain to grow efficiently on cellobiose. a** Schematic illustration of the application of ylAC for the optimization of cellobiose phosphorolysis pathway in *Y. lipolytica*. **b** Reserve carbon source glycogen content and PGM activities in phosphorolytic yeast (ylCello, and ylCello expressing different PGMs) compared with WT strain in aerobic growth on 10 g/L cellobiose. The presented data correspond to samples collected from the exponential growth phase. **c** Growth rate and residual cellobiose of yeast strains after 60 h of cultivation. Results were calculated from at least five biological replicates and are given as the mean value and standard deviation.

linear DNA fragments, unlike those of most gene integration sites in native chromosomes. Higher HR rates of gene integration have been achieved using site-directed nucleases, such as TALEN[16] and CRISPR/Cas9 or Cas12a/Cpf1[15,18,51,52], to introduce double-strand breaks (DSB) thus making the target integration site on the genome more accessible. Nevertheless, gene integration in natural chromosomes is highly dependent on the integration locus and on the size of homologous regions[28,29,39], two limits that are eliminated when using the ylAC.

Our results reveal that ARS and TEL are required to form complete, functional linear DNA assemblies. By recruiting protective protein caps[53,54], TEL may prevent the ends of chromosomes from fusing to one another, thus preventing the extremities from being recognized as double-strand breaks (DSB) and consequently avoiding HR and NHEJ[55].

It is known that repetitive sequences can promote recombination and cause instability of artificial chromosomes[6,49,56–58]. To avoid such behaviour, we characterized and used different promoters and terminators in ylAC. This should also facilitate and potentially allow controlling protein expression and modulating the ratio of protein component in multi-enzymatic pathways. Additionally, to overcome the common issue of foreign DNA loss in microorganisms when grown under non-selective conditions[48], the ylAC assembly can be conferred with essential functions that ensure its stability during mitosis. This ability to confer high stability makes ylAC2 an ideal tool for the construction of strains that will be used in industrial processes that require such a trait. Nevertheless, the simpler ylAC prototype might also be useful when transient expression of cytotoxic enzymes (e.g. nuclease) is targeted and/or when progressive loss of the construct is desirable. It is also noteworthy that the maintenance of stability is not limited to the use of auxotrophic selection, since our results show that this can also be achieved, for example, by selecting for the use of a specific carbon source (e.g.

growth on xylose or cellobiose), thus illustrating the extreme versatility of the ylAC system.

Previous work performed in *S. cerevisiae*[59,60] and *Y lipolytica*[51] has revealed that expression of chromosome-integrated genes is highly locus dependant. In this regard, the use of ylAC as an autonomous chromosome, devoid of down regulation systems that are present on the natural chromosomes, offers an elegant solution to avoid interference of gene expression in engineered strains. Clearly, this should open avenues for metabolic engineering in *Y. lipolytica*. However, it is also likely that ylAC will be useful to study the impact of chromosomal environments on gene regulation, the complete freedom to select the elements that surround the transcriptional unit being the driver for this. Moreover, it is foreseeable that the ylAC will be used for the precise reconstruction of DNA regulatory elements, modulatory sequences, identification of novel biobricks and evaluation of their efficiency in multi-enzymatic pathways, etc.

In conclusion, ylAC is a powerful genetic tool that holds enormous potential for researchers working on *Y. lipolytica*. As an orthogonal system, independent of native genome, and compatible with automation, ylAC, can be used for a variety of applications, including fundamental molecular genetics studies, enzyme discovery and engineering, and to perform synthetic biology. For more applied research, ylAC is transferable and should prove to be an invaluable addition to the metabolic engineer's toolbox, particularly for the development of industrially relevant *Y. lipolytica* strains.

## Methods

**Strains and media**. The genotypes of the microbial strains used in the present study are summarized in Supplementary Table 4. *E. coli* DH5α was purchased from Invitrogen (Paisley, UK) and used for plasmid construction. The *Y. lipolytica* strains were routinely cultured in a medium composed of 1% w/v yeast extract, 1% w/v Bacto peptone, and 1% w/v glucose (YPD). Transformants were selected on solid YNB medium (0.17% w/v YNB, 1% glucose or xylose or cellobiose w/v, 0.5% w/v ammonium chloride and 50 mM sodium-potassium phosphate buffer, pH 6.8),

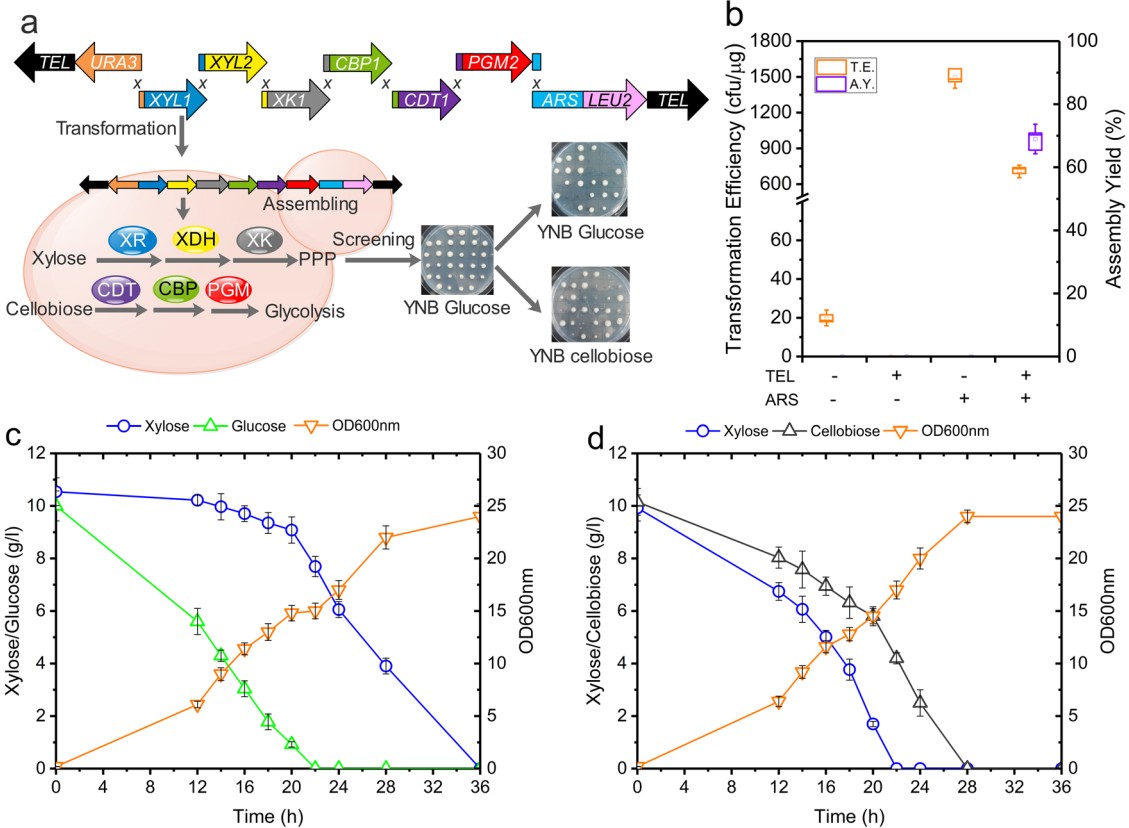

**Fig. 5 Assembly of 23 kb ylAC system. a** Schematic representation of the organization and construction of the ylAC (23 kb) bearing a cellobiose phosphorolysis and xylose consumption pathways. **b** Evaluation of TE and AY. **c** Growth curve of the recombinant strain ASXC1 in minimal media containing xylose and glucose. **d** Growth curve of the recombinant strain ASXC1 in minimal media containing xylose and cellobiose. Results were calculated from at least three biological replicates and are given as the mean value and standard deviation.

supplemented with uracil (100 mg/L) or leucine (200 mg/L) depending on the auxotrophic requirements. The detection of endoglucanase activity in solid YPD medium was achieved by incorporating 0.2% Azo-CMCellulose (Azo-CMC). For cellobiose phosphorylase and phosphoglucomutase activity assay, enzymes were produced in YTD medium (1% w/v yeast extract, 2% w/v tryptone, 5% w/v glucose and 100 mM phosphate buffer, pH 6.8). To compare the efficiency of cellobiose utilization with respect to cell growth, yeast was aerobically cultivated in minimal medium[61] containing vitamins, trace elements and salts, including 3.5 g/L $(NH_4)_2SO_4$, 3.0 g/L $K_2HPO_4$, 3.0 g/L $NaH_2PO_4$ and 1.0 g/L $MgSO_4 \cdot 7H_2O$ with 5 g/L (microplate) or 10 g/L cellobiose (shake flasks).

**Construction of a Y. lipolytica model telomere.** The telomeric repeats of Y. lipolytica are represented by an array of 10 bp 5'-TTAGTCAGGG-3' sequences whose lengths range from 500 to 1000 bp[62,63]. It has been suggested that the conserved 5'-TNAGGG-3' motif is important for the binding of telomere-associated protein Tay1p (YALIOD10923g), the only telomere-binding protein (TBP) found in Y. lipolytica so far[63]. In this work, a plasmid carrying the Y. lipolytica model telomere consisting of 63 5'-TTAGTCAGGG-3' repeats was constructed as previously described[63] with the following modifications (Supplementary Figs. 16 and 17). First, oligonucleotides ylTELF and ylTELR, containing nine telomeric repeats flanked with BamHI/BspMI on one side and HindIII/BbsI on the other (synthesized by Eurogentec, Belgium), were mixed in a 1:1 molar ratio and annealed. The resulting double-strand DNA (dsDNA) was digested with BamHI and HindIII (New England Biolabs) and cloned into the corresponding sites of plasmid pUC19, yielding plasmid pUC19-Tel Model. After that, the DNA fragment was recovered from the plasmid pUC19-Tel Model by BspMI/BbsI digestion. To extend the telomeric repeats, the resulting fragment was ligated into the plasmid pUC19-Tel Model digested with BspMI. After three rounds of BspMI digestion and ligation, the strategy yielded plasmid pUC19-Tel630 carrying 63 telomeric repeats.

**Construction of ylAC shuttle vector for multiple-gene expression.** The plasmids used in the present study are summarized in Supplementary Table 5, and all primers are listed in Supplementary Data 1. The ylAC shuttle vector should contain one telomere repeats combined with URA3 gene as selection marker and another telomere repeats combined with an ARS and LEU2 gene as selection marker. Some particular restriction enzyme recognition sites were placed to allow the easily recovery of the wanted modules. The LEU2 gene encoding beta-isopropylmalate

dehydrogenase was obtained by PCR using the plasmid JMP62LeuTEF[64] as template with primers LEF1 and LER1. The URA3 gene encoding orotidine-5'-phosphate decarboxylase was amplified from the plasmid JMP62UraTEF[64] using primers URAF1/URAR1. The ARS containing an origin of replication (ori3018) and a point centromere (CEN3-1) was amplified from the replicative plasmid pU18[65] of Y. lipolytica using primers ARF1/ARR1. The KpnI/NotI digested telomere and the BamHI/NotI digested telomere (both from pUC19-Tel630), the BamHI/EcoRI digested LEU2 gene, the KpnI/ClaI digested URA3 gene, and the ClaI/EcoRI digested ARS were then ligated into the plasmid pZK derived from plasmid JMP62UraTEF, yielding shuttle vector containing Y. lipolytica artificial chromosome (ylAC) (Supplementary Fig. 18).

**ylAC mediated multiple-gene assembling in Y. lipolytica.** The construction of the plasmids for the expression of reporter genes RedStar2, EG2, and of cellobiose phosprololysis pathway and xylose consumption pathways are described in Supplementary Methods. Using the constructed plasmids as template, PCR amplifications of the expression cassettes were carried out and 50 bp homologous regions were introduced into the PCR products to ensure the subsequent assembly.

(i)  Proof of concept: amplification of three expression modules with reporter genes
The expression cassette URA- pEXP-EGII-tLIP2 was amplified from the plasmid JMP62UraTB2EE2[18] by PCR using primers asEGF (containing the homologous sequence to the 5' end of URA3 gene) and asEGR. To assemble with pEXP-EGII-tLIP2 (upstream) and ARS (downstream), the expression cassette pTEF-RedStar2-tLIP2 was amplified from the plasmid JMP62LEU2expTEF-RedStar2 by PCR using primers asRedF1 (containing the homologous sequence to the expression cassette pEXP-EGII-tLIP2) and asRedR1 (containing the homologous sequence to the 5' end of ARS). To amplify the fragment pTEF-RedStar2-tLIP2 for assembling with pEXP-EGII-tLIP2 (upstream) and LEU2 gene (downstream), ssRedR2 flanking with homologous sequence to the 5' end of LEU2 gene was used as the reverse primer instead of asRedR1.

(ii) Amplification of three expression modules for cellobiose assimilation pathway
The expression cassette pTEF-CBP1-tLIP2 was PCR amplified from the plasmid JMP62UraTCbp1Tcyc using primers asCBPF1 (containing the homologous sequence to the 3' end of URA3 gene) and asCBPR. The

expression cassette *php4d-CDT1-tLIP2* was PCR amplified from the plasmid JMP62UrahCdt1 using primers asCDTF (containing the homologous sequence to the 3′ end of *pTEF-CBP1-tCYC1* and asCDTR. The expression cassette *pGUT-PGMs-tGUT* was PCR amplified from the plasmids pZP-GUT-PGMs (*ylPGM1, ylPGM2, ylPCM1, scPGM1* and *scPGM2*) using primers asPGMF and asPGMR (containing the homologous sequence to the 5′ end of *php4d-CDT1-Tlip2* and to the 3′ end of ARS, respectively.)

To assemble the cellobiose assimilation pathway with the xylose consumption pathway, the expression cassette *pTEF-CBP1-tLIP2* was PCR amplified from the plasmid JMP62UraTCbp1Tcyc using asCBPF2 as forward primer instead of asCBPF1, since it contains the homologous sequence to the 3′ end of *pFBA-XKS1-tFBA*.

(iii) Amplification of three expression modules for xylose consumption pathway.

The expression cassette *pTDH-XYL1-tTDH* was PCR amplified from the plasmid pZP-TDH-XR using primers asXRF (containing the homologous sequence to the 3′ end of *URA3* gene) and asXRR. The expression cassette *pEXP-XYL2-tEXP* was PCR amplified from the plasmid pZP-EXP-XDH using primers asXDHF (containing the homologous sequence to the 3' end of PCR fragment *pTDH-XYL1-tTDH*) and asXDHR. The expression cassette *pFBA-XKS1-tFBA* was PCR amplified from the plasmid pZP-FBA-XKS using the primers asXKSF (containing the homologous sequence to the 3′ end of PCR fragment *pEXP-XYL2-tEXP*) and asXKSR.

(iii) In vivo assembly

For *Y. lipolytica* transformation, ylAC was digested using *Not*I and *Cla*I, thus generating two linear fragments with telomere sequences at both extremities, each fragment containing a different selection marker (*URA3* on the left arm and *LEU2* fused with ARS on the right arm). These two linear DNA fragments were isolated by gel extraction and mixed with the target expression modules. Then the mixture was introduced into *Y. lipolytica* Po1d using the lithium acetate method[9] (Fig. 1).

For the proof-of-concept, three sets of experiments were performed to assemble the target pathways: (i) *URA3* gene without telomere (digesting ylAC by *Kpn*I and *Cla*I) and *LEU2* gene fused with ARS without telomere (digesting ylAC by *Bam*HI and *Cla*I); (ii) *URA3* gene fused with telomere (digesting ylAC by *Not*I and *Cla*I) and LEU2 gene fused with telomere without ARS (digesting ylAC by *Eco*RI and *Not*I); (iii) *URA3* gene (digesting ylAC by *Kpn*I and *Cla*I) and *LEU2* gene (digesting ylAC by *Eco*RI and *Bam*HI) directly (Fig. 2). For each assembling, transformation of the yeast cells without any DNA or with the target genes but without selection markers were used as the negative controls. Transformants were selected on YNB glucose plates. Transformation efficiency was calculated by dividing the number of successful transformants by the amount of DNA used during a transformation procedure.

To quantify the assembly yield, transformants expressing DsRed and EG2 reporter proteins were further tested for fluorescence production (DsRed activity), and EG2 activity assay on YPD plate containing Azo-CMC. Clones displaying both activities were retained for further analysis. In case of xylose or cellobiose consumption pathway assembling, transformants were further selected on YNB xylose or YNB cellobiose plates. The successful assembly of multiple genes in *Y. lipolytica* was verified by PCR using gene specific primers followed by DNA sequencing. Assembly yield was calculated by dividing the number of transformants harbouring the correct assemblies by the total number of the transformants verified.

**Determination of the copy numbers of the artificial chromosome.** To create a standard curve of gene copy numbers of *RedStar2* against Cq values, a real-time quantitative PCR (qPCR) was carried out using primers qRedf/qRedr from serial dilutions of *RedStar2* template gene. In parallel, a qPCR was performed using the above primers from the genomic DNA of *Y. lipolytica* harboring DNA assemblies of ylAC- *EG2-RedStar2* (ylAC-ER). Cells were grown in YNB media and collected at the exponential phase. The copy numbers of the *RedStar2* gene in the genome of *Y. lipolytica* was determined by comparing the Cq values of the standard curve using the obtained Cq values of the samples. Similarly, the copy numbers of the *ACT1* gene, encoding actin present at one copy number in the genome of *Y. lipolytica*, was determined and used as the reference for comparison.

**Measurement of enzyme activity.** Yeasts were cultivated until a cellular density of $6 \times 10^7$ cells/mL. A 50 mL sample was taken and subjected to centrifugation at $8000 \times g$ for 5 min at 4 °C thus isolating cell pellet and supernatant. The cell pellet was disrupted in Tris-HCl buffer (50 mM, pH 7.4, 3 mM EDTA and 0.5 mM PMSF) using a MP FastPrep-24 Instrument (MP Biomedicals Inc.). The activity of EG2 was measured on CMC (Megazyme) using a previously described method with slight modifications[50]. Briefly, the reaction mixture contained 1% (w/v) cellulosic substrate, 50 mM citrate buffer (pH 4.8) and a proper volume of diluted supernatant. The reaction was conducted at 50°C for 30 min and then the reducing sugars were quantified using the dinitrosalicylic acid (DNS) reagent. One unit of activity (U) was defined as the amount of enzyme required to release 1 μmol of the reducing sugars per min. Cellobiose phosphorylase activity in cell extract was assayed by measuring the formation of Glc-1P from cellobiose as described previously[66]. Briefly, the reaction mixture contained 10 mM cellobiose, 20 mM citrate buffer (pH 6.0) and a proper volume of diluted supernatant. The reaction was conducted at 60 °C for 10 min and then Glc-1P was determined using an enzymatic kit (G1P Colorimetric Assay Kit, Sigma), following the supplier's instructions. One

unit of activity (U) was defined as the amount of enzyme required to release 1 μmol Glc-1P per min. Phosphoglucomutase activity in cell extract was determined by a phosphoglucomutase activity assay kit following the supplier's instructions (Abcam, France). One unit of phosphoglucomutase is defined as the amount of enzyme that is needed to generate 1.0 μmol of NADH per min at pH 8 at room temperature. All protein concentrations were measured using the Bradford method and bovine serum albumin as a standard[67].

**Measurement of fluorescence.** Yeast was cultivated in minimal medium or YPD in shake flasks, at 28 °C, under shaking at 140 rpm, until stationary phase. Cell samples were taken regularly from the flasks and after being properly diluted (OD value ≈ 0.3), subjected to fluorescence assay using a Tecan infinite m200 pro spectrophotometer (Tecan Group Ltd., Switzerland). Samples were excited at 554 nm and fluorescence emission (around 591 nm) was measured. Alternatively, a Leica DM 4000B microscope was used to capture phase contrast and fluorescence images at ×100 oil immersion magnification.

**Transformants screening for cellobiose assimilation pathway optimization.** A single colony from fresh YPD plate was transferred into 5 mL of minimal medium containing 10 g/L of glucose or cellobiose and pre-cultured until the mid-exponential phase. The cells were then harvested, washed and suspended in sterilized water to inoculate a 48-well microplate of 200 μL minimal media containing 5 g/L cellobiose with an initial $OD_{600}$ of 0.05. This culture was grown in a microplate reader (Spectrostar Omega, BMG Labtech, Germany) at 30°C with continuous shaking (150 rpm) and automatic $OD_{600}$ recording.

**Analysis of glucose, cellobiose and intracellular metabolites.** Transformants showing different growth patterns on cellobiose in microplate were selected and further cultivated in 50 mL of minimal medium containing 10 g/L of cellobiose or glucose in shake flasks at 28 °C, shaking at 140 rpm. To determine the concentration of cellobiose and glucose in supernatants, three aliquots (1.5 mL each) of cultures were rapidly frozen in liquid nitrogen and then thawed on ice before centrifugation ($8000 \times g$ for 5 min at 4 °C) to recover supernatants for analysis. Glucose and cellobiose were separated and quantified using an Aminex HPX87-H column (Bio-Rad Laboratories, Germany) operating at 50 C with a mobile phase (5 mM $H_2SO_4$) flowing at a rate of 0.5 mL/min, and detected using a Shodex RI-101 refractive index detector (Showa Denko, New York, NY).

To measure the intracellular concentrations of Glc-1P, glycogen and trehalose, aliquots (1.5 mL each) of the above cultures at the exponential phase were immediately frozen using liquid nitrogen and stored at −80 °C. Before analysis, samples were thawed on ice. Cells were collected by centrifugation ($8000 \times g$ for 5 min at 4°C) and disrupted using glass beads as described above. Cellular Glc-1P was determined using an enzymatic kit (G1P Colorimetric Assay Kit, Sigma), following the supplier's instructions. Cellular glycogen and trehalose concentrations were measured as described previously[68]. Briefly, cells (~10 mg dry cell mass) were suspended in 250 μL 250 mM $Na_2CO_3$ and incubated at 95 °C for 4 h. After that, 150 μL of 1 M acetic acid and 600 μL of 0.2 M sodium acetate were added. For trehalose analysis, trehalase (0.05 U/mL) was added into the cell suspension and the reaction was conducted at 37°C for 2 h. For glycogen quantification, amyloglucosidase (1.2 U/mL) was added and the reaction was conducted at 57 °C for 2 h. Free glucose was quantified using the D-Fructose/D-Glucose Assay Kit according to the supplier's instructions.

**Determination of dry cell weight.** To determine the dry cell weight, three aliquots (5 mL each) of cultures were filtered through pre-weighed PES filters (0.45 μm; Sartorius Biolab, Germany). The biomass retained by the filters was washed, dried in a microwave oven at 150 W for 15 min, and then placed in a desiccator before weighting. The biomass yield was calculated as the ratio of the amount of biomass obtained divided by the amount of carbon source consumed.

**Southern blot analysis.** Southern blotting of the target DNA was performed as described previously[69] with the following modifications. Briefly, the gDNA of *Y. lipolytica* was digested by appropriate restriction enzymes, separated by gel electrophoresis and then transferred from the gel to a SensiBLot plus Nylon membrane (Thermo Fisher Scientific) by capillary action using absorbent paper to soak buffer solution through the gel and the membrane. Subsequently, the specific DNA fragments on the membrane were hybridized with the DNA probes prepared using the biotin decalabel DNA labeling kit (Thermo Fisher Scientific) and then detected by a biotin chromogenic detection kit according to the manufacturer's instructions.

**Determination of the stability of the DNA assemblies.** The stability of the linear and circular DNA assemblies in mitotic cells were determined as described previously[48]. Briefly, transformants harboring circular or linear ylAC-*EG* -*RedStar2* assemblies (ylAC-ER) were cultivated at 30 °C in YNB liquid media with and without supplementation of uracil and leucine until the mid-exponential growth phase. *Y. lipolytica* strains with the genes (without telomeric sequences) randomly integrated into their genome were used as the control for comparison. Then, cultures were transferred to the same fresh medium and this process was repeated

to maintain the cells in an exponential growth phase for 3 days. During the cultivations, cells were regularly taken and spread on YNB and YNB-ura-leu plate to determine the percentage of cells containing the DNA assemblies. The means and SDs of several determinations of the percentage of cells with circular or linear DNA assemblies are presented. The percentage of cells having lost their circular or linear DNA assemblies per generation (X) was determined as follows: $X = 1 - e^r$, where $r = \ln (A/B)/N$ (with $N$: number of generations; $A$ and $B$: percentage of cells containing circular or linear DNA assemblies at $N$ generations and at the beginning of the culture, respectively). The fraction of cells containing DNA assembly was determined as the number of cells grown on YNB plate divided by the number of cells grown on YNB-ura-leu plate.

**HEM1 gene deletion by CRISPR/Cas9**. HEM1 gene was deleted using CRISPR/Cas9 technology as described previously[17]. Single guide RNA (sgRNA) were designed using the online tool CRISPOR[70], which indicated the two sequences AAACGGGCGACTGACCCAGA (from 80 to 100 of ORF) and GGTGGGGCCGTACATACCAA (from 998 to 1017 of ORF) as potential suitable target sites. The former sequence was incorporated in a generic pCg58 plasmid (i.e. a plasmid bearing a Cas9 gene under the control of the TEF promoter and a sgRNA gene under the control of an hybrid ScR1-tRNAgly promoter, with URA3 as a marker of selection), while the latter was included in a pg7 generic plasmid similar to the previous one, but devoid of the Cas9 gene and with LEU2 as a marker of selection. The Po1d strain was co-transformed with both plasmids and transformants were selected on YNB plates supplemented with 100 μg/mL of 5-aminolevulinic acid (5-ALA). The deletion of HEM1 gene in the selected transformants was verified by PCR using primers ylHEM1F/ylHEM1R, followed by DNA sequencing.

**Optimization of ylAC by addition of HEM1 gene as an optional selection marker**. The DNA fragment of HEM1 gene locus, containing 746 bp upstream, 500 bp downstream and the ORF of HEM1 gene encoding 5-aminolevulinic acid synthase in Y. lipolytica, was amplified from gDNA of Y. lipolytica Po1d by PCR using primers ylHEM1F and ylHEM1R. Then, this DNA fragment was digested by I-sceI/ClaI and inserted into the corresponding sites of the plasmid ylAC, creating plasmid ylAC2. For Y. lipolytica transformation, ylAC2 was digested by NotI, I-sceI and ClaI. These three linear DNA fragments were isolated by gel extraction and mixed with the target expression modules before being introduced into Y. lipolytica Poldh. The transformants were selected on YNB plates supplemented with 100 μg/mL of 5-aminolevulinic acid (5-ALA). The successful assembly of target genes in Y. lipolytica was verified by PCR using gene specific primers followed by DNA sequencing.

**Statistics and reproducibility**. All the measurements were performed in at least three independent biological repeats unless otherwise stated. All source data underlying the main figures are available in Supplementary Data 2. Number of repeats (n) is indicated in figure legends and table notes, and Supplementary Data 2. Results are given as the mean value ± standard deviation. The results were analysed by two-tailed Student's t-tests, wherein P values ≤ 0.05 were considered statistically significant.

**Reporting summary**. Further information on research design is available in the Nature Research Reporting Summary linked to this article.

## Data availability
The source data underlying the Figures, Supplementary Figures, and Tables are provided in Supplementary Data 2. All the other data supporting the findings of this study are available within the article and its Supplementary Information Files and from the corresponding author upon reasonable request. The following plasmids will be available on Addgene.org (ylAC: 141182; JMP62UrahCdt1: 141214; JMP62UraTCbp1: 141215; pZP-EXP-DsRed: 141216; pZP-FBA-DsRed: 141217; pZP-GUT-DsRed: 141218; pZP-TDH-DsRed: 141219; pZP-EXP-XDH: 141220; pZP-FBA-XKS: 141221; pZP-GUT; scPGM2: 141222; pZP-TDH-XR: 141223).

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

## Acknowledgements

This work was performed using internal financial resources and we thank the catalysis and enzyme molecular engineering team of Toulouse Biotechnology Institute for the financial support. The authors would like to express their gratitude to Nelly Monties for her help with chromatographic analyses. We thank Prof. Jean-Marc Nicaud for providing some of the expressing vectors in Y. lipolytica, Myriam Mercade and Pascale Lepercq (Technical platform of quantitative and functional ecology) for their help in performing qPCR. We particularly thank Dr. Michael J. O'Donohue (CEPIA, Division of Science for Food and Bioproducts Engineering), Prof. Magali Remaud-Simeon (CIMEs) and Dr. Fayza Daboussi (Synthetic biology in microalgae) for constant discussion and valuable input.

## Author contributions

Z-P.G., C.C., G.T., S.D., A.M. and F.B. defined the concept; Z-P.G. designed the constructs, carried out the experiments and drafted the paper. V.B. contributed to the gene deletion using CRISPR/Cas9. Z-P.G., V.B., C.C., G.T., S.D., A.M. and F.B. revised the paper. All authors read and approved the final paper.

## Competing interests

The authors declare competing interests.
