## [Peer Review File · Communications Biology]

Reviewers' comments:

Reviewer #1 (Remarks to the Author):

The authors reported the development of artificial chromosomes as basic synthetic biology tools for the unconventional yeast *Y. lipolytica*. The work is innovative and holds significant promise for biorefinery applications that further advance *Y. lipolytica* as a promising chassis strain for industrial applications. A number of genetic elements, including ARS, telomere and centromere were functionally assembled with three unique genetic markers (*Ura3*, *leu2* and *Hem1*) to construct the yeast artificial chromosome yIAC. Fluorescence reporter gene and two carbon-utilization pathways were used to quantitatively understand the transformation efficiency and assembly yield.

A revision is required before this work could be considered for publication on Communications Biology.

- (1) For the centromere used in this study, could the authors specify whether this is a regional centromere (spanning across a few Kbs) or a site centromere (a few hundred bps)?
- (2) With bioinformatic tools and published genome database, Please illustrate and annotate the telomeric protein binding sites for the telomeres used in this yIAC.
- (3) The function of the spacer sequence between ARS and CENs and replication origin (ORI) may be impacted by epigenetic modifications. What would happen if the authors alter the orientation of ARS, CENs or use inverted repeats of CENs or with high GC spacers?
- (4) A complete set of literature review is important for readers to understand the landscape and innovation of the reported work. In the introduction part, the authors neglected to mention a number of synBio toolboxes recently developed, including self-replicative plasmids YaliBricks (published on Metabolic engineering communications), iterative gene integration and marker curation (ACS SynBio) and genome editing based on CRISPR-cpf1 (Metabolic Engineering Communications) et al. Please have this literature included or briefly discussed.
- (5) Page 5, line 107-109 "The transformants expressing RedStar2 under the control of different promoters and terminators showed similar growth rates and substrate consumption rates on YPD media". Please clarify whether this RedStar reporter is site-specific integrated onto the chromosome, if yes, where did the author integrate this RedStar. Based on which principle the authors did the author choose the integration site?
- (6) *Y. lipolytica* is known to have very strong autofluorescence, due to the accumulation of lipid bodies. Could the authors provide flow-cytometry images of the RedStar-expressing strain and control strain? Normally, a MFI (mean fluorescence intensity) is preferred to report the promoter strength.
- (7) Page 5, Line 110, "After glucose depletion, the fluorescence intensity of the cell suspensions was measured." It is preferred to measure the reporter activity at log phase, instead of the stationary phase of cell growth. Please clarify why the authors measured fluorescence after glucose is depleted.
- (8) To evaluate the Episomal status of the artificial chromosome, the authors performed a Southern Blot analysis. Since the yIAC is less than 10 kb, each of the native chromosomes is ranging from 1Mb to 4 Mb. It is very likely that this yIAC is integrated, or fused with the native chromosome. How could the author rule out this hypothesis? Could the authors perform more rigorous assay, like karyotype analysis with pulsed-field gel electrophoresis (PFGE) with authentic genome samples from the transformed cell?

(9) Is this yIAC a linear chromosome or a circular chromosome? What are the factors determining the copy number of this artificial chromosome?

(10) "in the absence of auxotrophic selective pressure, 184 both linear and circular yIAC gene assemblies were lost at substantially higher rates". This result indicates that the stability of yIAC is still auxotrophic-marker dependent. This may limit the transferability of the developed technique. It should be noted that HEM1 is still an auxotrophic marker.

(11) To test the functionality of the cellobiose and xylose pathway, is it possible that the authors test it with minimal media with cellobiose or xylose as sole carbon sources?

(12) For in vivo assembly of the gene cluster and the yIAC based on NHEJ, what is the size of the homologous arm used in this work?

Reviewer #2 (Remarks to the Author):

This is a reviewers report concerning the manuscript entitled "An artificial chromosome enables the efficient construction and expression of metabolic pathways in *Yarrowia lipolytica*", by Zhong-peng Gou et al. In this work the authors describe the construction of an yeast artificial chromosome-based platform for the first time in *Y. lipolytica*, and demonstrate its use for constructing an in vivo-assembled artificial cellobiose utilization pathway.

Overall, I think the work is well designed, the work is well performed with suitable controls, and the work is novel. Finally, I think it is of interest to the readers of Communications Biology. However, I have some concerns listed below. If the manuscript is adjusted to account for these, my recommendation is that the work is suitable for publication.

1) In figure 2 (and later figures), the transformation efficiency is contrasted with the assembly yield. The transformation efficiency is given as cfu/ug DNA. For some combinations (for instance TEL+, ARS- in fig 2b), very low numbers of colonies are achieved (around 10). However, no negative control is included here (i.e. cells transformed with no DNA). Thus I wonder what the number of colonies would be in this case, zero or higher? This crucial control should be added to the transformation efficiency experiments.

2) On page 5, it is stated that "The transformants expressing RedStar2 under the control of different promoters and terminators showed similar growth rates and substrate consumption rates on YPD media" with a reference to Figure S4. However, this figure does not show this, and only contains a single (representative?) growth curve.

3) In Fig S5, the experiments are inconsistently labeled compared with the manuscript. For instance, what is elsewhere labeled as "ARS- TEL+" is here only labeled with TEL+.

4) Overall, I think the figure text is quite small in the proof printout. Consider increasing this were possible.

Reviewer reports:

Reviewer #1:

Summary: The authors reported the development of artificial chromosomes as basic synthetic biology tools for the unconventional yeast *Y. lipolytica*. The work is innovative and holds significant promise for biorefinery applications that further advance *Y. lipolytica* as a promising chassis strain for industrial applications. A number of genetic elements, including ARS, telomere and centromere were functionally assembled with three unique genetic markers (Ura3, leu2 and Hem1) to construct the yeast artificial chromosome yIAC. Fluorescence reporter gene and two carbon-utilization pathways were used to quantitatively understand the transformation efficiency and assembly yield. A revision is required before this work could be considered for publication on Communications Biology.

Comments:

(1) For the centromere used in this study, could the authors specify whether this is a regional centromere (spanning across a few Kbs) or a site centromere (a few hundred bps)?

The centromere used in this study is a point centromere. We have specified this and added a schematic map to indicate the essential components of Ori and CEN of ARS in our revised manuscript (Page 17, Line 380-381; Supplementary figure S3).

(2) With bioinformatic tools and published genome database, please illustrate and annotate the telomeric protein binding sites for the telomeres used in this yIAC.

*It has been suggested that the conserved 5'-TNAGGG-3' motif within the telomere sequence is important for the binding of telomere-associated protein Tay1p (YALIOD10923g). In fact, Tay1p is the only telomere-binding protein (TBP) found in *Y. lipolytica* so far (Červenák, Cell Cycle, 2017). This information was added in our revised manuscript (Page 16, Line 355-358).*

(3) The function of the spacer sequence between ARS and CENs and replication origin (ORI) may be impacted by epigenetic modifications. What would happen if the authors alter the orientation of ARS, CENs or use inverted repeats of CENs or with high GC spacers?

*This comment is highly appreciated. The sequential originations of ARSs in the yeast *Y. lipolytica* has been studied previously (Fournier et al., PNAS. 1993; Vernis et al., MCB. 1997; Vernis et al., JMB. 2001). It has been shown that ARSs in *Y. lipolytica* require an origin of replication and a centromere, both of which are necessary for the maintenance of extrachromosomal DNA. In this work, we used the native ARS18 isolated from *Y. lipolytica* genome. ARS18 contains the origin 3018 located close to a centromere (CEN3-1). We did not attempt to alter the sequence of the inter linker between ORI and CEN since it has been shown that this tight, strict linkage sequence between the ORI and CEN has the scaffold-associated region (SAR) activity to bind to nuclear matrix, which is absolutely necessary for the extrachromosomal DNA maintenance. CEN3-1 is*

composed of juxtaposition of conserved sequences (CDEI box, A+T-rich sequence, CDEIII) and dyad symmetries. As the current CEN structural organization fulfill our purpose to develop a functional artificial chromosome, we did not further exploit the relationships of centromere structure and its function. Based on the published data, our assumption is that the modification of the current ARS organizations, such as insertion, deletion, partial inversion especially with high GC spacers (since ARS is A+T-rich) may cause dysfunctionality of the ARS (Fournier et al., PNAS. 1993; Vernis et al., MCB. 1997; Vernis et al., JMB. 2001). In the future, an independent study of detailed analysis of Y. lipolytica CEN sequences will be conducted to provide evidence for the relationship between the DNA structural elements and centromere function in Y. lipolytica. In addition, different centromeres, such as point centromere, regional centromere and transition centromere between the two are of interest to be tested.

(4) A complete set of literature review is important for readers to understand the landscape and innovation of the reported work. In the introduction part, the authors neglected to mention a number of synBio toolboxes recently developed, including self-replicative plasmids YaliBricks (published on Metabolic engineering communications), iterative gene integration and marker curation (ACS SynBio) and genome editing based on CRISPR-cpf1 (Metabolic Engineering Communications) et al. Please have this literature included or briefly discussed.

The authors thank the referee for this suggestion. The recently developed genetic toolboxes for the yeast Y. lipolytica were cited in our revised manuscript (Page 3, Line 45-46; Page 4, Line 78-80; Page 13, Line 289).

(5) Page 5, line 107-109 "The transformants expressing RedStar2 under the control of different promoters and terminators showed similar growth rates and substrate consumption rates on YPD media". Please clarify whether this RedStar reporter is site-specific integrated onto the chromosome, if yes, where did the author integrate this RedStar. Based on which principle the authors did the author choose the integration site?

For the expression of RedStar2 under the control of different promoters and terminators, the vectors were digested using NotI, this will generate a linear DNA with Zeta sequences at both extremities of the expression cassette. This will allow a single copy of the linearized constructs integrated into the Zeta docking platform of Y. lipolytica JMY1212 Zeta strain by homologous recombination after transformation. Our previous studies have showed that the gene integrated into zeta docking platform is constant and stable, thus allowing a better comparison of the expression levels of the target genes (Bordes et al. J Microbiol Methods, 2007). Moreover, we did fluorescence assays for at least five biological replicates of each construct and we confirmed the stability of the transformants with respect to RedStar expression. This has been clarified in our revised manuscript (Supplementary Table 1; Supplementary file 3).

(6) Y. lipolytica is known to have very strong autofluorescence, due to the accumulation of lipid bodies. Could the authors provide flow-cytometry images of the RedStar-expressing strain and

control strain? Normally, a MFI (mean fluorescence intensity) is preferred to report the promoter strength.

*We agree with the referee that when we use Bodipy (493 nm for excitation wavelength and 503 nm) as the fluorescent dye to stain lipid body of *Y. lipolytica*, this yeast exhibited strong autofluorescence. However, *Y. lipolytica* shows very low autofluorescence in the range used for the DsRed detection (554 nm for excitation wavelength and 591 nm), in fact it is completely indistinguishable from background at this wavelength by the fluorimeter. In this case, we have provided a detailed results of fluorescence production during the cell culture of *Y. lipolytica* expressing DsRed and used the wild type strain as the control (Supplementary figure S4). Our results revealed that average values for the fluorescence of a strain devoid of the gene for the fluorescent protein is 8 ± 3 A.U., the same obtained for wells filled with deionized water. On the other hand, cultures of the strain that carries a single copy of the redStar2 gene under the TEF promoter inserted in an intergenic region on the F chromosome, grown in parallel have a fluorescence of 630 ± 20 A.U. (Cell cultures were diluted so the Abs600 in the well were between 0.2 and 0.3, values obtained sampling a total of 16 cultures). Flow cytometry image, and correlation between fluorimeter measurements and flow cytometer data are available in the supplementary data of a previous report (Borsenberger, V. et al. Multiple Parameters Drive the Efficiency of CRISPR/Cas9-Induced Gene Modifications in *Yarrowia lipolytica*. *J. Mol. Biol.* 430, 4293-4306 (2018)).*

(7) Page 5, Line 110, "After glucose depletion, the fluorescence intensity of the cell suspensions was measured." It is preferred to measure the reporter activity at log phase, instead of the stationary phase of cell growth. Please clarify why the authors measured fluorescence after glucose is depleted.

Indeed, this is a mistake. As indicated in Page 22, Section "Measurement of fluorescence. Yeast was cultivated in minimal medium or YPD in shake flasks, at 28°C, under shaking at 140 rpm, until stationary phase. Cell samples were taken regularly from the flasks and after being properly diluted.....", we did follow the production of fluorescence of the cells during the whole cultivation process until the stationary phase when the maximum fluorescence intensity was achieved (after glucose depletion). In fact, although the fluorescence intensity of the cells varies during the cell cultures, the relative strength of the promoters/terminators remain constant. As this information could be useful for some readers, we have added these data in our revised manuscript (Supplementary figure S4)

(8) To evaluate the Episomal status of the artificial chromosome, the authors performed a Southern Blot analysis. Since the yIAC is less than 10 kb, each of the native chromosomes is ranging from 1Mb to 4 Mb. It is very likely that this yIAC is integrated, or fused with the native chromosome. How could the author rule out this hypothesis? Could the authors perform more rigorous assay, like karyotype analysis with pulsed-field gel electrophoresis (PFGE) with authentic genome samples from the transformed cell?

*The authors thank the referee for this suggestion. The integration or fusion of ylAC with the native chromosome can be ruled out by our carefully designed Southern Blot analysis. As indicated in Figure 3b and Supplementary figure S16b, when the untreated genomic DNA samples were analyzed by DNA electrophoresis, we were only able to detect one DNA band corresponding to the right size of the linear artificial chromosome using specific probes targeting ylAC. These results illustrated the ylAC is standing alone at episomal status, otherwise, the DNA band would show much higher molecular weight since the native chromosome is pretty large. When we digest the genomic DNA isolated from the yeast, we were only able to detect the DNA bands corresponding to the right sizes of the expected DNA fragments. These results indicated that the ylAC is not attached with other sequences otherwise different patterns of fragment distribution would be shown after restriction enzyme digestion. Furthermore, we were able to isolate ylAC and re-transform it into the yeast *Y. lipolytica*. And further isolate the genomic DNA and the Southern Blot analysis of the isolated ylAC after second transformation again confirmed the episomal status of the ylAC. Since similar assays have been performed on different samples (at least three biological replicates) and for different constructs (independent experiments), we believe these data are sufficient and reliable for obtaining the conclusion of the episomal status of the ylAC. Indeed, we have tried to visualize the ylAC with other techniques such as DNA-FISH. However, similar to the problem of karyotype analysis, the artificial chromosome was too small to be detected. In this case, PFGE is not necessary as the current electrophoresis fulfilled our requirement of sufficient resolutions for the detection of the ylAC in our Southern Blot analysis.*

(9) Is this ylAC a linear chromosome or a circular chromosome? What are the factors determining the copy number of this artificial chromosome?

*In fact, ylAC can exist in both linear or circular form. However, when we increase the number of the target genes (≥ 3) to be assembled with ylAC, we only observed linear form of chromosome (Supplementary file 2, Figure S14, S16). The copy numbers of ylAC is mainly determined by ARS sequence. Our results and the others' illustrated that ARS18 plasmids displayed low copy numbers (1 to 3) and higher mitotic stability, which are different from that of traditional ARS elements in *S. cerevisiae*. These unusual properties were attributed to the presence of a centromere within the ARS elements which restores a balanced segregation between mother and daughter cells and a 2:2 meiotic segregation of ylAC (Fournier, et al., PNAS, 1993).*

(10) "in the absence of auxotrophic selective pressure, 184 both linear and circular ylAC gene assemblies were lost at substantially higher rates". This result indicates that the stability of ylAC is still auxotrophic-marker dependent. This may limit the transferability of the developed technique. It should be noted that HEM1 is still an auxotrophic marker.

We agree with the referee that selective pressure is necessary for the maintenance of ylACs, even the HEM1 gene functions similarly to an auxotrophic marker. However, we do believe that these are not limiting factors which would prevent the wide usage of the developed ylACs in metabolic

engineering of Y. lipolytica. First of all, minimal media, which are the most industrially relevant culture media, provide the auxotrophic selective pressure. Second, in case of the necessity to grow the yeast in rich media, HEM1 can be used as the selection marker since most of media does not contain 5-ALA as nutrients. Particularly, with the recently developed genome edition tools, such as Crispr/Cas9 or Cas12a/Cpf1, it is much easier to develop such 'auxotrophic' selection markers. In any case, there is no need to further remove (rich media) or supplement with additional nutrients (minimal media) for the cell cultivations. Moreover, researchers are completely free to introduce their own selection markers or methods, which can be an auxotrophic marker, an essential gene, or a pathway for utilization of a specific carbon source, etc., which depends on the genes or pathways assembled on the yLAC.

(11) To test the functionality of the cellobiose and xylose pathway, is it possible that the authors test it with minimal media with cellobiose or xylose as sole carbon sources?

Minimal media with cellobiose or xylose as sole carbon sources were used in our study to grow the yeast. Maybe we did not clarify this enough as we used YNB media for the transformants selection but we monitored the growth and consumption of cellobiose or xylose in minimal media. In any case, these results were clarified in our revised manuscript (Page 15, Line 340-344; Page 16, Line 348-352; Page 22, Line 498-500, 503-506; Legend of Figure 5).

(12) For in vivo assembly of the gene cluster and the yLAC based on NHEJ, what is the size of the homologous arm used in this work?

We have used 50-bp short homologues arms. Again, this is an advantage of the yLAC developed in this study, as 50-bp homologues sequence can be easily introduced by normal PCR with proper primer design. This information has been further clarified in our revised manuscript (Page 17-18, Line 391-393; Legend of Figure 2).

Reviewer #2:

Summary: This is a reviewers report concerning the manuscript entitled “An artificial chromosome enables the efficient construction and expression of metabolic pathways in *Yarrowia lipolytica*”, by Zhong-peng Gou et al. In this work the authors describe the construction of an yeast artificial chromosome-based platform for the first time in *Y. lipolytica*, and demonstrate its use for constructing an in vivo-assembled artificial cellobiose utilization pathway.

Overall, I think the work is well designed, the work is well performed with suitable controls, and the work is novel. Finally, I think it is of interest to the readers of Communications Biology. However, I have some concerns listed below. If the manuscript is adjusted to account for these, my recommendation is that the work is suitable for publication.

Comment:

1) In figure 2 (and later figures), the transformation efficiency is contrasted with the assembly yield. The transformation efficiency is given as cfu/ug DNA. For some combinations (for instance TEL+, ARS- in fig 2b), very low numbers of colonies are achieved (around 10). However, no negative control is included here (i.e. cells transformed with no DNA). Thus I wonder what the number of colonies would be in this case, zero or higher? This crucial control should be added to the transformation efficiency experiments.

As suggested by the reviewer, we have performed the control experiments for all the different constructs. No colonies were observed when the yeast was transformed without any DNA or with the target genes but without the selection markers. These results were added in our revised manuscript (Page 7, Line 137-139; Page 20, Line 442-444). In fact, the very low transformation efficiency of TEL+, ARS- is likely due to the presence of the telomeric sequences attached to the selection makers URA3 and LEU2, although the underlying mechanism remains to be elucidated.

2) On page 5, it is stated that “The transformants expressing RedStar2 under the control of different promoters and terminators showed similar growth rates and substrate consumption rates on YPD media” with a reference to Figure S4. However, this figure does not show this, and only contains a single (representative?) growth curve.

Since the transformants expressing RedStar2 under the control of different promoters and terminators showed similar growth rates and substrate consumption rates on YPD media, we only showed a representative growth curve for all. This has been clarified in the legend of Figure S4a.

3) In Fig S5, the experiments are inconsistently labeled compared with the manuscript. For instance, what is elsewhere labeled as “ARS- TEL+” is here only labeled with TEL+.

We have modified all the labeled and keep them constant through the manuscript (Figure S5).

4) Overall, I think the figure text is quite small in the proof printout. Consider increasing this where possible.

Following the reviewer's advice, we have increased the size of the text in all the figures.

REVIEWERS' COMMENTS:

Reviewer #1 (Remarks to the Author):

The authors have successfully addressed the concerns. I suggest the acceptance of this article with its current form.